 elife.elifesciences.org

# Changing the responses of cortical neurons from sub- to suprathreshold using single spikes in vivo

Verena Pawlak[1], David S Greenberg[1], Henning Sprekeler[2], Wulfram Gerstner[2,3], Jason ND Kerr[1,3]*

[1]Network Imaging Group, Max Planck Institute for Biological Cybernetics, Tübingen, Germany; [2]School of Computer and Communication Sciences and School of Life Sciences, Brain Mind Institute, Ecole Polytechnique Federale de Lausanne, Lausanne, Switzerland; [3]Bernstein Center for Computational Neuroscience, Tübingen, Germany

**Abstract** Action Potential (APs) patterns of sensory cortex neurons encode a variety of stimulus features, but how can a neuron change the feature to which it responds? Here, we show that in vivo a spike-timing-dependent plasticity (STDP) protocol—consisting of pairing a postsynaptic AP with visually driven presynaptic inputs—modifies a neurons' AP-response in a bidirectional way that depends on the relative AP-timing during pairing. Whereas postsynaptic APs repeatedly following presynaptic activation can convert subthreshold into suprathreshold responses, APs repeatedly preceding presynaptic activation reduce AP responses to visual stimulation. These changes were paralleled by restructuring of the neurons response to surround stimulus locations and membrane-potential time-course. Computational simulations could reproduce the observed subthreshold voltage changes only when presynaptic temporal jitter was included. Together this shows that STDP rules can modify output patterns of sensory neurons and the timing of single-APs plays a crucial role in sensory coding and plasticity.

*For correspondence:
jason@tuebingen.mpg.de

**Competing interests:**
The authors have declared that no competing interests exist

**Reviewing editor:**
Michael Häusser, University College London, United Kingdom

## Introduction

The formation of sensory percepts in the mammalian cortex is thought to involve the propagation of action potentials through populations of connected neurons in response to sensory stimulation (*Hebb, 1949*; *Harris, 2005*). Rules that dictate changes in the strength, or efficacy, of these connections rely on persistent correlated activity between the connected neurons (for recent review, see *Abraham, 2008*). For these changes in synaptic efficacy to be functional and recognized by the down-stream neurons targeted by the axon, the changes need to influence AP patterns. Plasticity induction involving precisely timed action-potentials, known as spike-timing-dependent plasticity (*Markram et al., 1997*; *Bi and Poo, 1998*), has been suggested as a candidate mechanism to determine which stimulus driven inputs are strengthened and weakened (out of the many possible sensory driven inputs). What makes STDP rules so appealing for modification of synaptic strength in the cortex is that the low numbers of APs and low repetition frequencies used to induce plastic changes in STDP protocols (*Feldman, 2012*) are consistent with low AP firing rates observed in awake freely moving animals exploring a novel environment (*Lee et al., 2006*; *Sawinski et al., 2009*). In addition, cortical neurons are capable of AP-responses that are temporally precise over multiple trials (*Bair and Koch, 1996*; *Marsalek et al., 1997*; *Tchumatchenko et al., 2011*), similar to the precision typically used during STDP pairing protocols (for recent review see, *Feldman, 2012*). Lastly, most modeling studies of synaptic plasticity using STDP rules assume that synaptic modification leads to changes in spiking (*Gerstner et al., 1996*; *Song and Abbott, 2001*) with a recent study suggesting that STDP underlies sensory map plasticity in the

**eLife digest** Nerve cells, called neurons, are one of the core components of the brain and form complex networks by connecting to other neurons via long, thin 'wire-like' processes called axons. Axons can extend across the brain, enabling neurons to form connections—or synapses—with thousands of others. It is through these complex networks that incoming information from sensory organs, such as the eye, is propagated through the brain and encoded.

The basic unit of communication between neurons is the action potential, often called a 'spike', which propagates along the network of axons and, through a chemical process at synapses, communicates with the postsynaptic neurons that the axon is connected to. These action potentials excite the neuron that they arrive at, and this excitatory process can generate a new action potential that then propagates along the axon to excite additional target neurons. In the visual areas of the cortex, neurons respond with action potentials when they 'recognize' a particular feature in a scene—a process called tuning. How a neuron becomes tuned to certain features in the world and not to others is unclear, as are the rules that enable a neuron to change what it is tuned to. What is clear, however, is that to understand this process is to understand the basis of sensory perception.

Memory storage and formation is thought to occur at synapses. The efficiency of signal transmission between neurons can increase or decrease over time, and this process is often referred to as synaptic plasticity. But for these synaptic changes to be transmitted to target neurons, the changes must alter the number of action potentials. Although it has been shown in vitro that the efficiency of synaptic transmission—that is the strength of the synapse—can be altered by changing the order in which the pre- and postsynaptic cells are activated (referred to as 'Spike-timing-dependent plasticity'), this has never been shown to have an effect on the number of action potentials generated in a single neuron in vivo. It is therefore unknown whether this process is functionally relevant.

Now Pawlak et al. report that spike-timing-dependent plasticity in the visual cortex of anaesthetized rats can change the spiking of neurons in the visual cortex. They used a visual stimulus (a bar flashed up for half a second) to activate a presynaptic cell, and triggered a single action potential in the postsynaptic cell a very short time later. By repeatedly activating the cells in this way, they increased the strength of the synaptic connection between the two neurons. After a small number of these pairing activations, presenting the visual stimulus alone to the presynaptic cell was enough to trigger an action potential (a suprathreshold response) in the postsynaptic neuron—even though this was not the case prior to the pairing.

This study shows that timing rules known to change the strength of synaptic connections—and proposed to underlie learning and memory—have functional relevance in vivo, and that the timing of single action potentials can change the functional status of a cortical neuron.

young and adult cortex (*Young et al., 2007*). However, evidence for single-neuron STDP in vivo remains scarce and confined to subthreshold changes (*Meliza and Dan, 2006*; *Jacob et al., 2007*) and remains controversial (*Lisman and Spruston, 2005, 2010*; *Fregnac et al., 2010*; *Shouval et al., 2010*). While extracellular studies have suggested STDP can modify spiking patterns in anesthetized (*Fu et al., 2002*) and freely moving animals (*Celikel et al., 2004*), this evidence remains difficult to interpret without information on single-cell membrane potential dynamics.

The visual cortex is perfectly suited to study STDP since visual cortex neuronal responses and receptive fields are known to be highly plastic (*Gilbert, 1998*), and sensory maps can undergo plastic changes with experience (*Daw and Wyatt, 1976*; *Li et al., 2008*) or after removal of visual input (*Wiesel and Hubel, 1963*; *Kaas et al., 1990*; *Gilbert and Wiesel, 1992*). Finally during visual stimulation, the action potential output for many visual cortex neurons is tuned to specific locations in the visual field and to specific stimulus features (*Hubel and Wiesel, 1959*), even though each neuron receives synaptic inputs corresponding to surrounding locations or many different stimulus features (*Ferster, 1986*; *Jia et al., 2010*). Together these conditions make a perfect system to test whether STDP rules can, with a few tens of spikes, reliably change a subthreshold responding neuron into a neuron responding to the same stimulus with APs and change a neuron's visually driven AP patterns in a timing dependent manner.

# Results

We addressed these questions by pairing precisely timed postsynaptic spikes to presynaptic inputs arising from visual stimulation. We used STDP protocols similar to those previously applied in vivo (*Meliza and Dan, 2006*), but with the difference that all ongoing membrane potential voltage fluctuations and spiking at the cell soma were measured and taken into account when presenting the stimulus, since large membrane potential fluctuations (*Cowan and Wilson, 1994*) can alter a neuron's spiking responses (*Petersen et al., 2003*; *Sachdev et al., 2004*) and ongoing spiking may influence plasticity outcome (*Sjostrom et al., 2001*). We targeted whole-cell recordings (*Margrie et al., 2003*) to Layer 2/3 (L2/3) neurons in the binocular primary visual cortex (n = 58 rats) using intrinsic imaging followed by two-photon microscopy as using stereotaxic coordinates alone can be inaccurate (*Slotnick and Brown, 1980*; *Kline and Reid, 1984*). All neurons (n = 71, yielding 72 data points, see 'Materials and methods') were identified as L2/3 pyramidal neurons, based on firing pattern, location below the pia and morphology after post hoc histological staining. For visual stimulation, a bar was flashed (500 ms) (*Figure 1A*) every 3–4 s at one of four non-overlapping spatial positions (*Figure 1B*). To measure ongoing spiking and membrane potential fluctuations, and to reduce variance and maximize consistency in the evoked responses (*Petersen et al., 2003*; *Sachdev et al., 2004*), we recorded in current clamp mode and always presented stimuli after the membrane potential had transitioned from an up-state into a down-state (*Figure 1C*), during which far fewer spontaneous synaptic inputs occur (*Waters and Helmchen 2006*).

All four stimulus positions evoked either subthreshold or suprathreshold responses, consistent with the broad receptive fields of neurons in younger rats (*Fagiolini et al., 1994*). Mean subthreshold responses reached an initial peak 50–150 ms after stimulus onset, then decayed to a trough within 100–200 ms (*Figure 1D*), though considerable variability (*Hirsch et al., 2002*) was observed in the peak time, response shape, and amplitude across neurons (*Figure 1D*). After a stable baseline period (9.0 ± 1.3 min, 32.4 ± 1.9 presentations of each stimulus, n = 32 recordings), we applied an STDP induction protocol by pairing single-APs, evoked by brief depolarizing current pulses, with visual stimulation at only one of the four stimulus positions (42.0 ± 1.5 pairing trials, 1.3 ± 0.16 APs/trial, n = 32, *Figure 1E*). Since each neuron's response time was different, the timing of the pairing spikes was individually chosen for each neuron to occur either before ('negative timing inductions', n = 17, example in *Figure 1E*, left) or after the initial peak ('positive timing inductions', n = 15, example in *Figure 1E*, right). As a result, by taking all recordings together based on whether the pairing spike was delivered before (red, negative timing) or after (green, positive timing) the initial peak, the time of pairing spikes ranged from 82 ms before the response peak to 62 ms after (*Figure 1F*). After pairing, we again recorded the neuron's responses to stimuli at each of the four positions without injecting current pulses (51.9 ± 3.6 presentations of each stimulus over 31.8 ± 2.4 min, starting 1.0 ± 0.4 min after the last paired stimulus, n = 32, *Figure 2A*). Responses at the paired position were changed, with the direction of the change strongly dependent on the whether the pairing spike appeared before or after the response peak (*Figure 2B,C*). As expected (for review see, *Bi and Poo, 2001*) in neurons that received negative timing inductions the peak voltage amplitude significantly decreased by 26.4 ± 4.6% (n = 17) or 3.5 ± 0.7 mV (*Figure 2C*, p<0.0001), while positive timing inductions increased the peak voltage amplitude by 22.5 ± 8.7 % (n = 15) or 1.4 ± 0.6 mV (*Figure 2C*, p=0.025). The difference between what was observed here compared to previous in vitro (*Bi and Poo, 2001*) and in vivo studies (*Fregnac et al., 2010*) was that although the direction of the plasticity observed in responses to visual stimulation depended on spike timing, the largest initial peak changes were not around t = 0 but were scattered from 0–50 ms on either side of t = 0.

The small changes in membrane potential observed here and in other studies raise the issue of whether the observed plasticity is sufficient to modify spiking responses in the postsynaptic neuron. Synaptic plasticity of a given neuron's inputs is functionally effective only if it alters neuronal spiking, so that information is passed on to downstream synaptic targets. To link STDP to changes in spiking at the single cell level, we therefore tested whether a sensory-driven but subthreshold-responding neuron (*Figure 2D*, left) could be changed through STDP induction into a suprathreshold-responding neuron (*Figure 2D*, right). In most neurons that received positive timing inductions sensory-evoked spiking rates increased after the pairing protocol (*Figure 2E*). Specifically, of the 15 neurons tested, 11 showed an increase in the average number of APs evoked by stimulus presentation, 1 showed a decrease, and 3 never discharged a stimulus-evoked AP (*Figure 2G*). Strikingly, in eight neurons the paired stimulus evoked spiking only after induction, that is over 50% transitioned from non-spiking to spiking. Over the 15 neurons tested, an additional 0.11 ± 0.06 APs per stimulus were observed for the

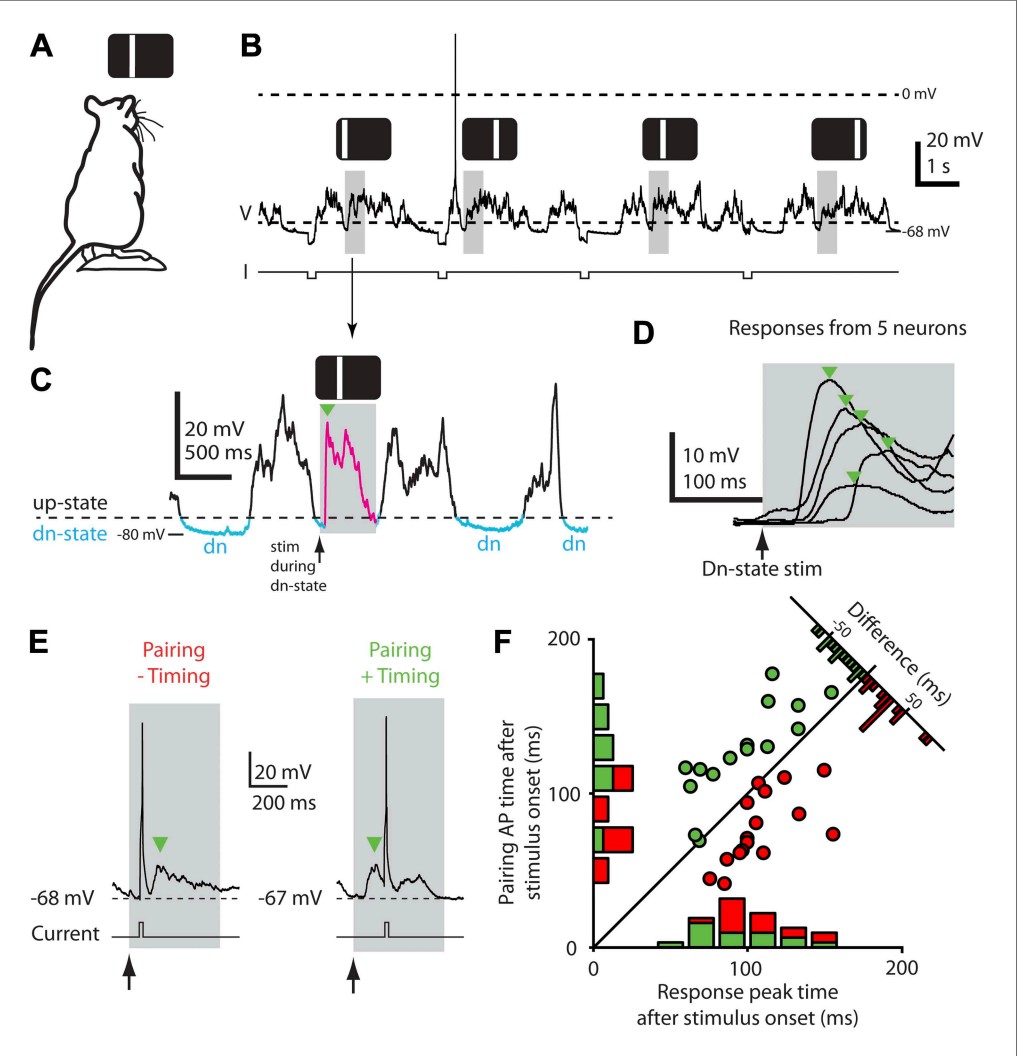

**Figure 1**. Down-state triggering of visual stimuli and timing of evoked responses and spikes during the STDP protocol. (**A**) Flashed vertical bars were presented to an anesthetized rat from a stimulus screen. (**B**) Continuous whole-cell current-clamp recording with down-state triggered visual stimulus presentations (grey boxes) and hyperpolarizing step currents (−100 pA) to estimate input resistance. Flashed vertical bars were presented in one of four different positions across the stimulus screen. (**C**) Detailed view of whole-cell current-clamp recording of a voltage response (magenta, green arrow marks initial peak) to a 500 ms (grey box) flashed bar presented only during down-states (period marked in blue, dashed line indicates threshold used for online down-state detection). (**D**) Average membrane potential responses (green arrows indicate initial response peaks) to flashed bar stimulus for five different neurons. (**E**) Pairing consisted of injecting depolarizing current into the postsynaptic neuron to elicit a single AP timed to occur either before (left, '−Timing') or after (right, '+Timing') the response peak (green arrow). (**F**) Scatter plot showing timing of evoked APs for each recording vs timing of initial response peak (n = 32). Time 0 denotes stimulus onset. Histograms show the number of neurons with positive timing and negative timing.

paired stimulus position after induction (p=0.01, initial response 0.06 ± 0.04 APs per stimulus). In four cases increases in AP firing were significant at the single-neuron level (p<0.05), despite the modest numbers of stimulus presentation trials and low firing rates (see 'Materials and methods').

We next tested whether the STDP protocol could decrease the sensory-evoked spiking rates using negative timing inductions, in which pairing spikes occurred before the initial peak. For this to occur, we identified a stimulus position that evoked spiking in the recorded neuron (successful in 15 out of 17 recordings, 0.84 ± 0.23 APs per stimulus), and then proceeded to pair spiking with stimulus presentation at this position. After pairing, the average number of APs evoked by stimulus presentation

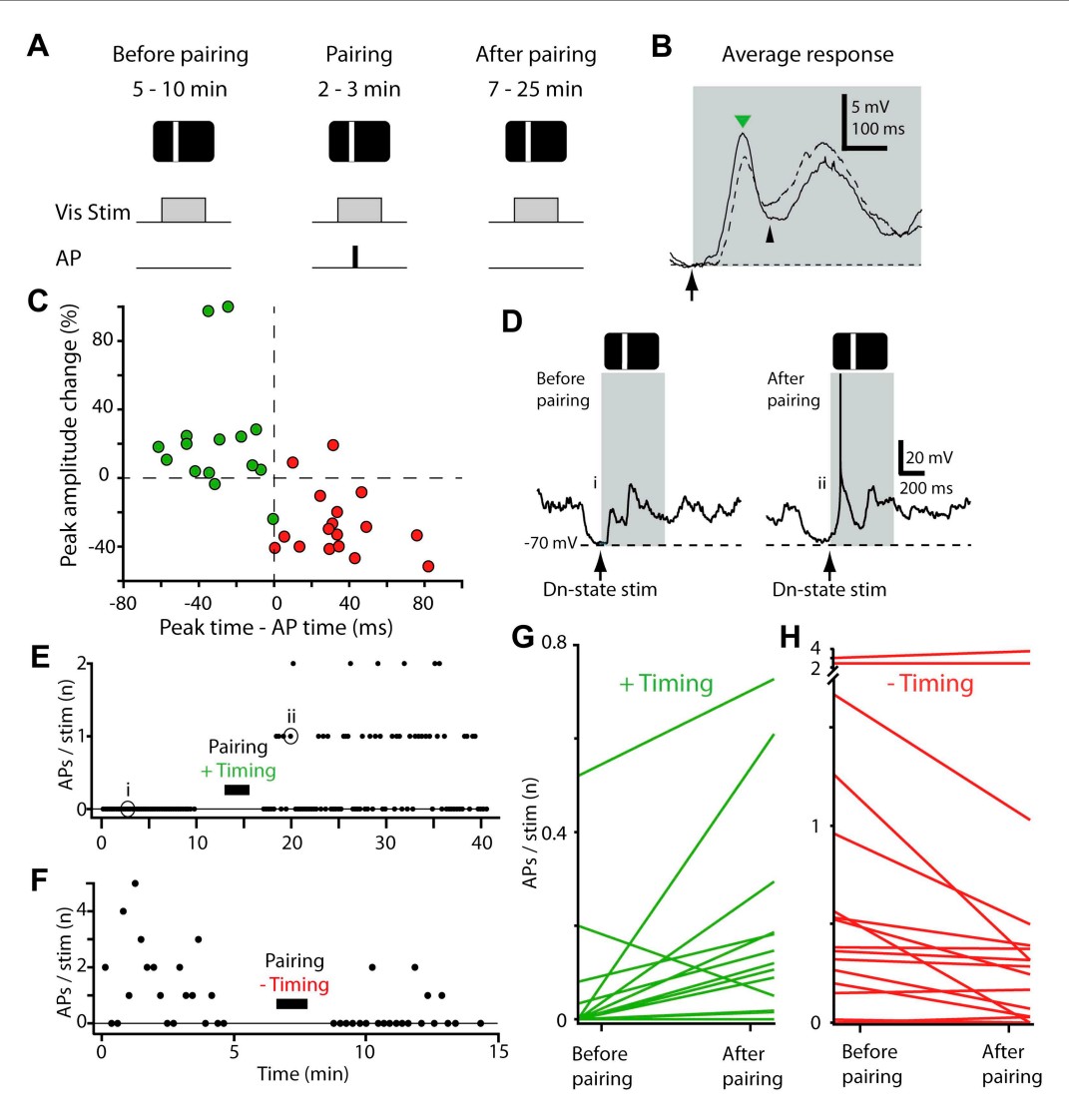

**Figure 2**. STDP converts subthreshold responses to visual stimuli into suprathreshold responses. (**A**) Time course of the experiment. Before the pairing, responses to visual stimulation were recorded, during the pairing, visual stimulation was paired with APs evoked by brief current injection (repeated 40 times), followed by recording of visual responses. (**B**) Average membrane potential responses to stimulus presentation at the pairing position, before (dashed line) and after pairing (black line). The portion of the response before the pairing APs are potentiated, while the portion after is depressed. Small black arrow indicates timing of pairing APs. (**C**) Scatter plot showing the change in peak amplitude vs the time difference between the pairing APs relative to the peak. Delivering APs before the peak (red circles) decreases the peak amplitude, while delivering them after increases it (green circles). (**D**) Subthreshold membrane potential response to flashed bar (grey box) before pairing with a postsynaptic spike after the initial peak (positive timing), and suprathreshold response to same flashed bar stimulus after pairing. (**E**) Number of APs evoked per visual stimulus before and after pairing, for the neuron shown in (**D**). (**F**) Number of APs evoked per visual stimulus before and after pairing, for a neuron that received a negative timing protocol (postsynaptic spike before the initial peak). (**G**) Average number of APs evoked per stimulus before pairing compared to after pairing for all neurons with positive timing induction (n = 15). (**H**) Average number of APs evoked per stimulus before pairing compared to after pairing for all neurons with negative timing induction (n = 17).

decreased (0.18 ± 0.10 APs per stimulus after induction, p=0.04, n = 15). Of these 15 neurons 12 showed a decrease in spiking and 3 an increase after induction (*Figure 2H*). These changes were statistically significant at the single-neuron level (p<0.05) in four cases, all of which showed a decrease in spiking. Together this shows that an STDP protocol that induces small bidirectional changes in the membrane potential can both increase and decrease stimulus-evoked spiking depending on the relative timing of the single pairing spikes.

These changes in spiking responses observed after STDP induction could not be accounted for by neurons becoming more depolarized (down-state membrane potential −64.8 ± 1.5 mV before vs −64.2± 1.5 mV after, p=0.61, n = 32) or by input resistance changes (323 ± 39 MΩ before vs 419 ± 54 MΩ after, p=0.16), and neither depolarization nor input resistance changes were correlated with changes in spiking (p>0.05). In a control group, where visual stimuli were presented but no APs were paired, stimulus-driven spiking rates in the second half of the recording were not significantly different from the first half (p=0.20, four neurons with four positions each), indicating that the increase in spiking was due to the STDP protocol. Finally, we note that increases in AP firing at the paired position for positive timing inductions remained significant even after removing the two neurons exhibiting the greatest increases (p=0.036, n = 13). Together, these results provide a robust demonstration that repeated pairing of a single AP to a visual stimulus ~40 times can initiate spiking responses in non-spiking neurons and can increase or diminish existing spiking responses. This is consistent with an STDP-driven change in functional status and provides experimental validation that STDP could provide a plasticity mechanism for changing receptive fields (*Young et al., 2007*).

We next measured whether induced spiking changes also extended within the receptive field to non-paired stimulus positions (*Figure 3A*). For positive timing experiments, we observed diverse changes in the spiking responses measured at non-paired positions as spiking elicited at positions directly adjacent to the paired position (±1) increased in some neurons (*Figure 3B–D*) but decreased in others (*Figure 3E*), and the position eliciting the maximum response often changed after pairing (*Figure 3D,E*). Compared to the average 0.11 ± 0.05 additional APs per stimulus evoked at the pairing position after induction, stimulus positions immediately adjacent to the pairing position also showed an increase nearly as large (0.10 ± 0.05 APs per stimulus, p=0.02, n = 15). This increase in AP firing diminished for more distant positions (*Figure 3G*) and was no longer significant for stimulus positions not immediately adjacent to the paired position (p>0.05). In negative timing experiments, changes in AP firing were not significant at any non-paired positions (p>0.05).

We further examined whether these changes displayed any common pattern, and might admit some simple description. To test this we considered neurons' spiking responses as functions of stimulus position, and calculated the 'stimulus response centers' of these functions in the space of stimulus positions (*Figure 3D,E*, dashed lines). Stimulus response centers were calculated as weighted averages of the four stimulus positions, with each position weighted by the rate of stimulus-evoked spiking. This analysis showed a simple but robust trend in the positive timing experiments: after stimulus-AP pairing, the stimulus response center moved toward the pairing position (*Figure 3F*). The shift in the suprathreshold response center was strongly correlated with the difference between pairing position and the response center before pairing (R = 0.81, p=0.008). For the negative timing inductions the majority of response centers moved away from the pairing position, but these changes were small and the trend was not significant (p=0.70).

We next examined whether the shifts in response centers observed for suprathreshold responses were also reflected in subthreshold responses. Subthreshold response centers, calculated by weighting each position by the peak amplitude, also shifted toward the pairing position for positive timings, albeit much less robustly than the suprathreshold shifts (R = 0.53, p=0.04, *Figure 4A–C*). As with the suprathreshold response centers, for negative timings subthreshold response centers tended to move away from the paired position, but these changes were too small to be significant (p>0.05, *Figure 4C*). This consistency between the suprathreshold and the subthreshold response center changes brought about by an STDP induction protocol shows that plasticity of stimulus responses was timing-specific and spatially structured, with transfer to neighboring regions of the receptive field. These results show that STDP protocols induce a rearrangement of the sub- and suprathreshold receptive field structure, especially for positive timing protocols.

A key difference between the present study and previous in vitro STDP experiments (*Bi and Poo, 1998*), where activation of presynaptic inputs is temporally precise and brief, is that in vivo during sensory stimulation cortical L2/3 neurons receive a more complex and longer-lasting synaptic input barrage (*Figures 1D, 2B*) with variable timing of individual inputs (*Fagiolini et al., 1994*; *Niell and*

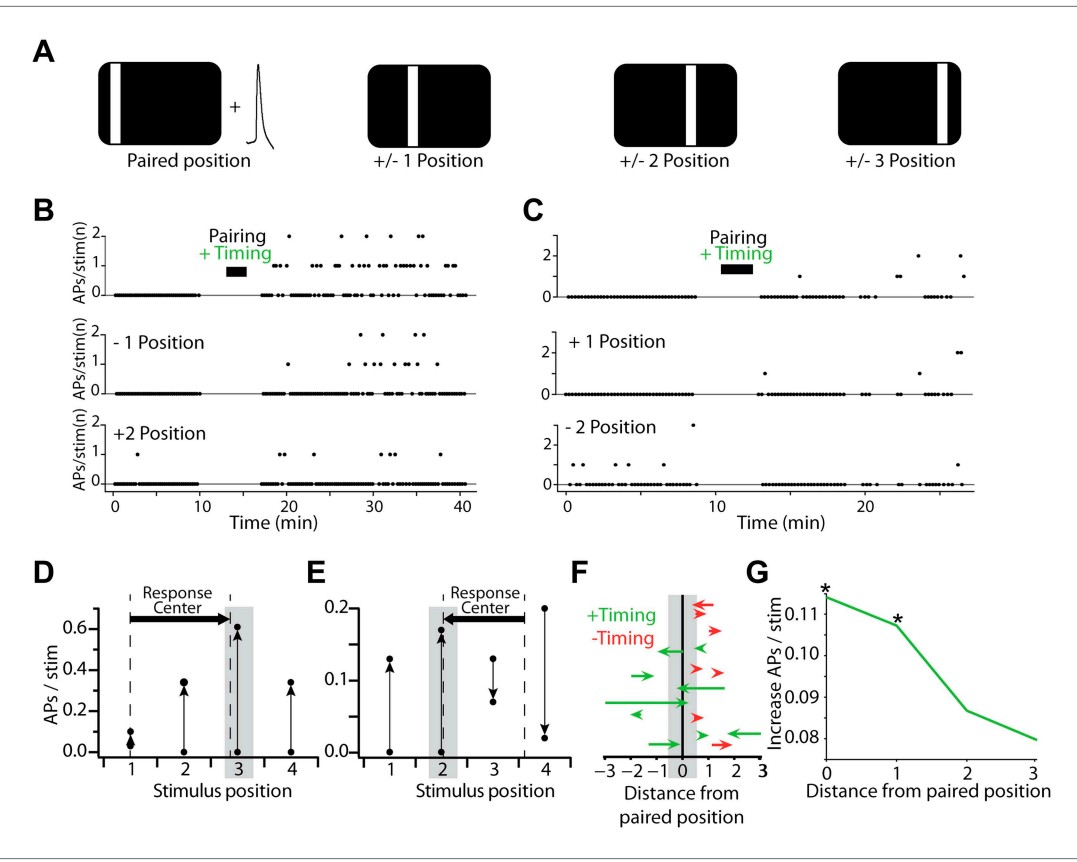

**Figure 3**. Changes in sensory-evoked spiking are stimulus-specific and lead to spatial reorganization of neurons' stimulus responses. (**A**) Stimulus-AP pairing was performed for a single stimulus position, but responses to neighboring and more distant positions were also measured before and after the pairing phase of the experiment. (**B**) Number of APs evoked per visual stimulus before and after pairing with positive timing. Stimulus-evoked spiking is shown for the paired position (upper), a neighboring position (center) and a more distant position (lower). The increase in stimulus-evoked spiking is greatest at the paired position, and the size of the change decays with distance. (**C**) As in (**B**), but showing an example where the paired position displays an increase in firing (upper) while a more distant position displays a decrease (lower). (**D**) Rates of spiking evoked by sensory stimulation in the recording shown in (**B**), before and after pairing (paired position marked by grey box); vertical arrows indicate direction of the change from before to after pairing. Dashed vertical lines indicate the suprathreshold stimulus response center before and after pairing, which shifted from position 1.00 to 2.86. (**E**) As in (**D**), but for the recording shown in (**C**), where a more distant position decreased its rate of stimulus-evoked spiking; response center shifted from 3.62 to 2.06. (**F**) Shifts in the suprathreshold stimulus response center after the pairing. Each arrow indicates the response center positions before and after pairing for a single experiment with positive (green) or negative timing (red). Vertical positions of arrows are arbitrary. Note that suprathreshold response center shifts could not be calculated for some recordings with insufficient spiking (see 'Materials and methods'). Positive timing experiments display a shift toward the paired position. (**G**) Increase in stimulus-evoked spiking for neurons receiving positive timing inductions, as a function of distance from the paired position. Asterisks indicate significance at the $p < 0.05$ level (Wilcoxon rank sum test).

*Stryker, 2008*). Despite this in vivo variability, traditional STDP rules dictate that inputs consistently arriving before the pairing spike should be potentiated and increase the response amplitude before the spike, while those arriving consistently after should be depressed and reduce the response amplitude after the spike. Finally, if each response is, on average, made up of the same synaptic inputs occurring in the same relative temporal order, then the stimulus-AP pairing should not simply change the peak amplitude, but rather induce a complex change that varies continuously over the time course of the subthreshold response relative to the pairing spike (*Figure 5A*, same data as *Figure 2B*). We therefore calculated the time course of changes in the subthreshold response relative to the time of the pairing spike for all neurons (positive and negative timing inductions, n = 32, *Figure 5B*, grey lines). Thus, in

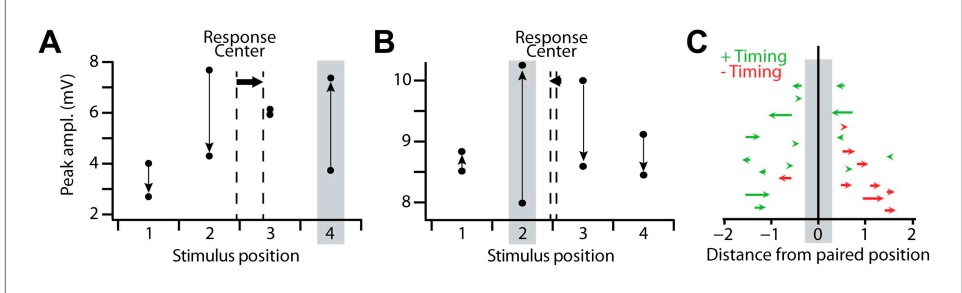

**Figure 4**. Spatial reorganization of subthreshold responses. (**A**) Mean amplitude of the subthreshold response peak in a single recording with a positive timing induction, for all stimulus positions before and after pairing. Grey box indicates the pairing position, vertical arrows indicate the direction of the change from before to after pairing, and dashed vertical lines the subthreshold stimulus response center before and after pairing, which shifted from position 2.44 to 2.89 (horizontal arrow). (**B**) As in (**A**), but for a recording in which subthreshold responses at the pairing position transitioned from weakest of the four positions to strongest after pairing; response center shifted from 2.55 to 2.46. (**C**) Shifts in the subthreshold stimulus response center after stimulus-AP pairing, for positive timing inductions (green) and negative timing inductions (red). Each arrow indicates the response center positions before and after pairing for a single experiment with positive (green) or negative timing (red). Vertical positions of arrows are arbitrary.

contrast to traditional STDP analysis where a neuron provides only a single data point on a graph of time difference vs response change (*Figure 2C*), both potentiation and depression occur for each neuron as a continuous function of time that on average is biphasic (*Figure 5B*, black line) with an increase before the spike (maximum increase 0.84 mV at 64.5 ms before, p=0.04) and a decrease thereafter (maximum decrease 1.4 mV at 43.5 ms after, p=0.01). This showed that STDP protocols in vivo can simultaneously induce potentiation and depression in the membrane potential of the same neuron (*Figure 2B*). Changes in membrane potential were strongest at the paired position, with similarly shaped but smaller changes at adjacent positions consistent with overlap in synaptic inputs, and nondescript changes at more distant positions (*Figure 5C*). While the shape of these membrane potential changes was qualitatively similar to the biphasic STDP curve observed in vitro (*Markram et al., 1997*; *Bi and Poo, 1998*), the positive and negative peaks were temporally further away from the pairing spike and shape of the curve were much broader in vivo.

We hypothesized that this peak shift and broadening effect results from variable timing of synaptic inputs from L4 and L2/3 (*Fagiolini et al., 1994*; *Niell and Stryker, 2008*), since potentiating or depressing an input with more variable jitter in the timing will potentially affect a greater range of the subthreshold stimulus response. To test this, we built an STDP model with a plasticity rule (*Gerstner et al., 1996*; *Song et al., 2000*) consistent with previous in vitro studies (*Figure 5D–K*) to examine STDP effects for various presynaptic activity patterns (*Figure 5E,H*). The membrane voltage in the neuron model is the sum of excitatory postsynaptic potentials (EPSPs), each described as a double exponential with rise time 2 ms and decay time 10 ms. Amplitudes of EPSPs were chosen from an exponential distribution with mean amplitude of 2 mV. With this set of parameters and 400 presynaptic inputs, the summed EPSPs had amplitudes and time courses similar to those measured in the experiment. Since the model was designed to describe experiments where spiking is induced artificially by the experimenter, firing is not an intrinsic property of the model but firing times were set 'by hand' at the target times designated for current injection by the experimental protocol (*Figure 5D*). Modeling each input's stimulus response as fixed and temporally precise produced membrane potential changes similar to those reported for in vitro STDP timing curves (*Figure 5F*). However, introducing jitter into the presynaptic activity (*Figure 5G*) caused the positive and negative peaks of the change in membrane potential to broaden and become more distant from the pairing spike with increasing jitter (*Figure 5G*). With a presynaptic jitter of 40 ms the model produced a biphasic change in the mean membrane potential (*Figure 5H,I*) that closely resembled the in vivo experimental results (*Figure 5B*), with a maximal potentiation 50 ms before and maximal depression 74 ms after the postsynaptic spike. The model also reproduced partial transfer of STDP effects to neighboring stimulus positions (*Figure 5J*, red) in a spike timing dependent manner (*Figure 5K*) and given the assumption that a subset of presynaptic neurons respond with similar spiking patterns for multiple stimulus positions. These modeling results

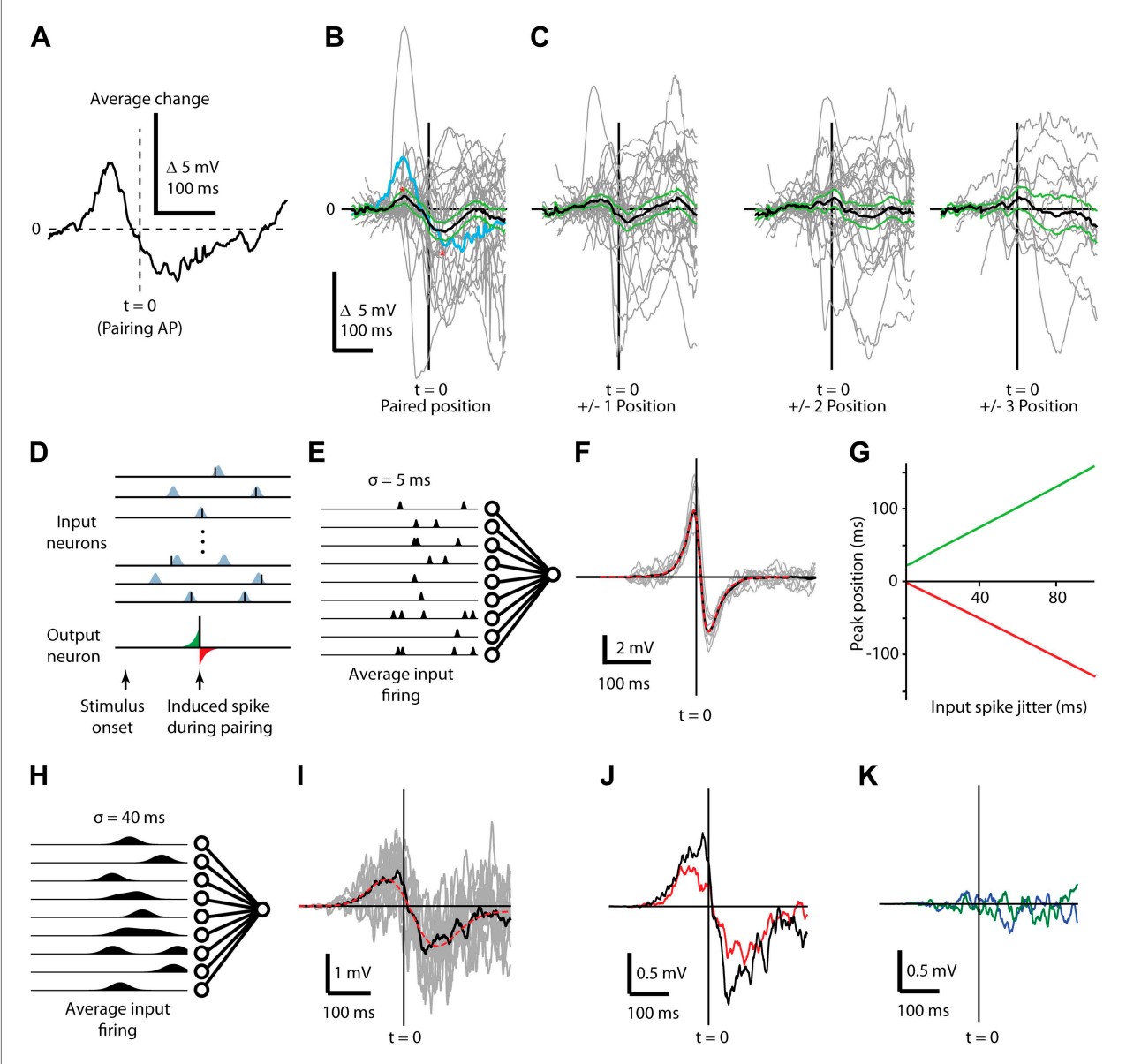

**Figure 5**. Changes in subthreshold responses after STDP induction display temporally broad peaks consistent with pre-synaptic jitter. (**A**) Average difference between before and after pairing membrane potential time course in response to visual stimulation at the paired position for a single neuron, aligned to the time of the pairing spike (dashed vertical line, t = 0). (**B**) Average (black) difference between before and after pairing membrane potential response to visual stimulation, as a function of time difference from the pairing spike. Individual neurons are shown in grey, green lines indicate SEM and the example neuron from (**A**) is shown in blue. (**C**) As in (**B**), but for non-paired stimulus positions adjacent to the paired position (left) and more distant positions (center, right). (**D**) Structure of the computational model showing that for each stimulus 400 presynaptic neurons were activated, providing input to a single postsynaptic neuron. Input spikes had a Gaussian probability distribution around target firing times. During pairing, a single postsynaptic spike (bottom) is induced at a predetermined time and input weights are increased if an input spike arrives at the respective synapse just before the induced spike (green area) and decreased if an input spike occurs just after the induced spike (red area). (**E**) Schematic showing mean firing rates over time for input neurons with σ = 5 ms used for calculating F. (**F**) Average difference in membrane potential when the input spike precision is ~5 ms (individual simulations grey, mean red) compared to a theoretical curve. (**G**) Timing of the positive (green) and negative (red) peaks of the membrane potential change relative to the induced postsynaptic spike as a function of the jitter in input spike timing. (**H**) Schematic showing mean firing rates over time for input neurons with σ = 40 ms used for calculating I. (**I**) Average difference in membrane potential when the input spike precision is ~40 ms. (**J**) Transfer of plasticity to a neighboring stimulus position when a subpopulation of synapses and timing are shared (red) compared to membrane potential difference from (**I**, black). (**K**) Transfer of plasticity to a neighboring stimulus position when a subpopulation of synapses are shared but timing is not (blue), and when no synapses are shared (green).

*Figure 5. Continued on next page*

*Figure 5. Continued*
The following figure supplements are available for figure 5:

**Figure supplement 1**. Input spike patterns for the computational model of in vivo STDP effects.

**Figure supplement 2**. Membrane potential responses in the postsynaptic output neuron for various activity patterns used to model the firing of presynaptic input neurons.

**Figure supplement 3**. Changes in membrane potential responses to stimulus presentation after pairing for all activity patterns used to model the firing of presynaptic input neurons.

show that the membrane potential changes observed in vivo are broader than those observed in vitro and that the broader time course can be explained by variably-timed inputs (but see *Lin et al., 2003*; *Froemke et al., 2005*; *Zhang et al., 2009* for alternative sources of broader STDP windows). Additionally, the slight transfer of changes in synaptic strength from one stimulus position to the next within a receptive field can be explained by partial overlap of the inputs activated by different stimuli as previously suggested (*Meliza and Dan, 2006*).

## Discussion

Our results demonstrate that an STDP protocol using less than 50 paired APs can convert a neuron's visually evoked responses from non-spiking to spiking, can increase and diminish existing spiking responses, and can induce a rearrangement of the neurons sub- and suprathreshold responses to stimuli presented at locations surrounding the paired location. Our current findings extend our knowledge about STDP-induced subthreshold changes in vivo (*Meliza and Dan, 2006*; *Jacob et al., 2007*) to the suprathreshold domain. Our study also provides a bridge between the changes in spiking observed across many neurons' receptives fields after alteration of sensory input for example through retinal lesions (*Wiesel and Hubel, 1963*; *Kaas et al., 1990*; *Gilbert and Wiesel, 1992*) and subtle changes in subthreshold responses induced by STDP at the single neuron level both in vitro (*Magee and Johnston, 1997*; *Markram et al., 1997*; *Bi and Poo, 1998*) and in vivo (*Meliza and Dan, 2006*; *Jacob et al., 2007*). Such a connection between receptive field plasticity, rearrangement of a neuron's tuning properties and STDP has been suggested by theoretical studies (*Song and Abbott, 2001*; *Young et al., 2007*), but to date has not been shown experimentally. As the rates of AP activity used in the pairing protocols was similar to activity reported for upper cortical layer populations in the visual cortex of both headfixed awake (*Greenberg et al., 2008*) and freely exploring rodents (*Sawinski et al., 2009*), this form of plasticity could represent a more general plasticity principle that relies on low firing rates and is central to individual neurons joining or leaving cell-assemblies, as proposed by Donald Hebb (*Hebb, 1949*).

In the present study we manipulated the timing of spikes relative to an avalanche of stimulus-triggered presynaptic events and observed significant changes in the firing properties of the postsynaptic neuron. In other words, we focused on manifestations of Hebbian plasticity that depended upon spike timing. We cannot exclude, however, that purely subthreshold manipulation of the postsynaptic voltage could have caused similar, or even stronger effects of Hebbian plasticity (*Lisman and Spruston, 2005*, *2010*; *Fregnac et al., 2010*; *Shouval et al., 2010*). While STDP highlights the timing-dependence on a millisecond time scale, Hebbian synaptic plasticity also manifests itself in experimental protocols manipulating firing rates and postsynaptic voltage (*Artola et al., 1990*; *Sjostrom et al., 2001*). In fact, previous models assuming that Hebbian synaptic plasticity depends critically on postsynaptic voltage, have allowed general theories of plasticity with both rate- and timing-dependent plasticity as special cases (*Clopath et al., 2010*). Optimality theories of STDP developed over the last 10 years, have proposed that the fundamental reason why pre-before-post induces LTP is to strengthen the causal link between input and output. Thus, the optimal timing between pre- and post should be related to the rise time or width of the PSP caused by the presynaptic input (*Pfister et al., 2006*; *Bohte and Mozer, 2007*; *Toyoizumi et al., 2007*; *Parra et al., 2009*; *Hennequin et al., 2010*; *Pool and Mato, 2011*). There is no similar optimality theory as to why synaptic depression should be spike-triggered, as unspecific forms of depression of synapses (not temporally linked to the output spike) would work just as well from a theoretical point of view (*Pfister et al., 2006*; *Bohte and Mozer, 2007*;

*Toyoizumi et al., 2007*; *Hennequin et al., 2010*; *Pool and Mato, 2011*). What our study proposes is that timing based post-before-pre LTD (negative timing) is a robust mechanism for manipulating neuronal spiking responses along with the previously proposed pre-before post LTP (positive timing) (*Gerstner et al., 1996*; *Song and Abbott, 2001*).

The present study shows a robust timing effect for the change in voltage of the initial peak, where negative timing protocols produced decreased synaptic efficacy and positive timing protocols resulted in an increased synaptic efficacy, similar to that seen in vitro. What is not consistent with previous in vitro STDP studies, where synaptic efficacy changes were derived from post-synaptic potentials, is our reported lack of dependence of the peak changes on the milisecond timescale. From in vitro studies it would be expected that a pre- before post pairing with a delay during pairing of around 40 ms would produce almost no change and a pre- before post with a delay of around 2 ms would produce the most change (*Bi and Poo, 1998*). Our reported initial peak changes were not at t = 0 but were seen for a diverse range of timings within a time range of ~100 ms (*Figure 2C*). We suggest that this discrepancy with previous studies (also see *Fregnac et al., 2010*) is due to the complexity of in vivo responses to visual stimulation which are made up of an avalanche of both inhibitory and excitatory synaptic events (*Hirsch et al., 1998*; *Okun and Lampl, 2008*). It is therefore an oversimplification to capture such complex responses by measuring only the peak of the initial response. Here we measured the evoked voltage across the entire time course of the response and aligned the voltage changes around the timing of pairing APs. With this approach we were able to show a very robust timing dependence, not just of an isolated synaptic input, but of the integrated inputs made up of multiple synaptic events (*Figure 5A–C*). The peak response of this voltage change was not always at t = 0 as would have been expected from in vitro experimental studies (*Markram et al., 1997*; *Bi and Poo, 1998*; *Froemke et al., 2005*) and assumed in STDP timing models. We found peak response changes at about 30–50 ms before and after the pairing spike. Our model suggests that this discrepancy is due to the very precise pre-synaptic timing during the pairing protocol from one trial to the next used in in vitro STDP studies which is contrary to the variable L4 and L2/3 visual response latencies found in vivo (*Fagiolini et al., 1994*; *Niell and Stryker, 2008*). In our STDP model, variability of input timing alone was sufficient to explain this broadening effect in our data compared to that expected from in vitro STDP studies.

The absolute voltage changes brought about by STDP in our study were on the order of 2 mV, yet such a change was sufficient to significantly alter the stimulus evoked spiking probability. Hence, our study suggests that small voltage changes have a dramatic effect on spiking output. We can only speculate on the mechanism because the dependence of spike generation on synaptically driven depolarizations is very complex (*Shu et al., 2007*), and somewhat unknown. This is especially the case in vivo (*Hasenstaub et al., 2007*) where combinations of various non-linear currents over the sub-threshold voltage range are present (*Waters and Helmchen, 2006*) and the spatial location of active synaptic inputs during evoked sensory stimulation (*Varga et al., 2011*) are not known. Despite these in vivo uncertainties, complex membrane-potential waveforms at the soma in vitro can reliably generate spikes (*Mainen and Sejnowski, 1995*) and recent simulations paired with experiments (*Tchumatchenko et al., 2011*) show that cortical pyramidal neurons can rapidly and reliably change firing rates in response to very small changes in somatic depolarization and hyperpolarization (twofold spiking difference induced with a 40 pA current injection).

In the current study we show that spike-timing-dependent plasticity rules using very few pairing APs can turn a neuron that hitherto did not emit a spike in response to a visual stimulus into one that frequently spikes, while pairing APs with a different timing can reduce established spiking. We propose that this is a fundamental mechanism by which neurons join and leave cell-assemblies (*Wallace and Kerr, 2010*) as previously proposed (*Hebb, 1949*) in which very few precisely timed spikes select specific temporal synaptic sequences for long-term storage from the many that are activated during a sensory learning process. Since these spikes will be transmitted to hundreds of other downstream neurons, such a change in tuning has direct implications for the function of neuronal circuits in which the respective neuron is embedded.

## Materials and methods

### Animal preparation

All experimental procedures were performed according to the animal welfare guidelines of the Max Planck Society. Male Lister Hooded rats at postnatal day 23–28 were anesthetized with urethane

(1.6–2 g/kg body weight, i.p.). Supplementary doses of urethane (10% of original dose) were administered as required to maintain anesthetic depth at a level where paw withdrawal and corneal reflexes were absent. The general anesthesia depth during all recordings were identified as stage III-3 based on lack of whisker movement and tail response, and the frequency spectrum of electrocorticogram (ECG) signals (*Kubicki and Rieger, 1968*; *Friedberg et al., 1999*). Body temperature was monitored throughout the experiment with a thermal probe and maintained at 37°C. The animal skull was exposed and cleaned and a metal plate was attached with dental acrylic cement. A craniotomy and a dural opening were made above the left cortex (2.5 mm anterior lambda, 4.2 mm lateral from the midline). The exposed cortex was superfused with Normal Rat Ringer's (NRR) solution (in mM): 135 NaCl, 5.4 KCl, 5 HEPES, 1 $MgCl_2$ and 1.8 $CaCl_2$ (pH 7.2) (*Kerr et al., 2005*). Astrocytes were labeled with sulforhodamine 101 (*Nimmerjahn et al., 2004*). The craniotomy was filled with agarose (type III-A, 1% in NRR) and covered with an immobilized glass coverslip.

## Intrinsic optical imaging

The animal's eyes were covered with cotton pads and NRR during the surgery until immediately before the imaging session. The part of primary visual cortex responding to visual stimuli presented directly in front of the animal's nose was determined using intrinsic optical imaging (*Grinvald et al., 1986*). The cortical surface was illuminated with red light (630-nm interference filter) while presenting moving gratings (see below) to both eyes. Reflectance images were acquired with a Cascade CCD camera (type 512B; Photometrics, Tucson, AZ, USA) equipped with a lens pair of a 135-mm and a 50-mm focal length objective (NA 0.46, 2.7× magnification) using custom software written in Labview (National Instruments, Austin, TX, USA). Briefly, individual trials were 6 s long (100 ms frame duration), with 2 s prestimulus time and 4 s stimulus time. Typically, 'blank' trials were alternated with stimulus presentation trials and a total of 40 trials comprised one imaging block. Imaging blocks were conducted for two different stimulus types: bars (white on black background, individual bar 2.4° wide) moving (at 11°/s) vertically and horizontally. Care was taken to shield the optical system from stimulus light. The surface blood vessel pattern was imaged for reference using green illumination (546 nm).

## Two-photon imaging

Two-photon imaging was performed using a custom-built two-photon laser-scanning microscope (excitation wavelength 872 nm, Ti:Sapphire laser model Mai Tai HP; Spectra-Physics, Newport, RI, USA). Fluorescence images were acquired as described before (*Kerr et al., 2005*).

## Electrophysiology

An ECG was recorded with the tip of a 500-µm-diameter Teflon-coated silver wire placed against the pial surface in one corner of the craniotomy. A reference electrode was placed over the cerebellum through a small hole in the occipital bone. Electrocorticogram signals were acquired with a custom-built AC-coupled amplifier (input impedance 1 MΩ, bandwidth 0.1 Hz to 8 kHz). The animal's eyes were covered with cotton pads soaked with NRR until shortly before establishing whole-cell recordings. To establish whole-cell configuration in vivo, neurons were visualized as dark spots in the sulforhodamine 101 counterstain using two-photon fluorescence imaging (*Kerr et al., 2005*) and approached with a patch pipette. In a few cases, in which for example surface blood vessels prevented sufficiently clear visualization of single neurons in the targeted area, the 'blind' patch technique (*Margrie et al., 2003*) was used to establish whole-cell configuration. Altogether, in vivo whole-cell recordings from 71 neurons (n = 58 rats) were established. In four recordings no STDP induction protocol was applied (see below), in 66 recordings exactly one STDP induction protocol was applied, and in one extraordinarily long recording (2 hr 17 min) two STDP induction protocols were applied (spaced more than 1 hr apart). Thus, the presented data set consists of n = 72 data points. Recorded neurons were located in primary visual cortex in L2/3, 169 ± 3 µm below the pial surface (range 112–225 µm, median = 170 µm; lower quartile = 150 µm; upper quartile = 186 µm) and within the responsive area identified with intrinsic optical imaging (see above, diameter of targeted area was approximately 850 µM around the center of the spot generated by intrinsic imaging). Since the location of this area was very consistent across animals in a set of initial experiments (n = 58 rats, with not all initial experiments yielding data points), intrinsic imaging was performed only in this initial set of experiments and in the following experiments, neurons were targeted stereotactically in this location. Only recordings from neurons that showed firing properties characteristic of pyramidal neurons were included, whereas fast-spiking neurons were excluded.

Patch pipettes (5–9 MΩ) contained (in mM): 135 K-gluconate, 10 HEPES, 10 phosphocreatine-Na, 4 KCl, 4 ATP-Mg, 0.3 GTP-Na, 0.02 Alexa Fluor 594, and 0.2% biocytin (pH 7.2, 310 mOsm). Membrane potential was recorded using an Axoclamp 2B amplifier (Molecular Devices, Sunnyvale, CA, USA). Input resistance was estimated by application of current steps (−100 pA, 200 ms). Data was digitized at 31.5 kHz using a CED Power 1401 data acquisition board (CED; Cambridge Electronic Design, Cambridge, United Kingdom).

## Visual stimuli during electrophysiology

Visual stimuli (500 ms flashed bars, white against a black background) were triggered by the membrane potential entering a down-state (*Kerr and Plenz, 2002*) (trigger for visual stimuli was typically ~60 ms from down-state onset). These flashed bars (horizontal or vertical bars) were presented in 1 of 4 alternating locations on an Iiyama (Vision Master Pro 21, Tokyo, Japan, refresh rate 60 kHz) CRT monitor (1600 × 1200 pixels) placed 34 cm away from the animal's nose spanning 60.9° in width and 48° in height of the visual field. Bars were non-overlapping and either 15.2° wide and 48° high (vertical bars) or 60.9° wide and 12° high (horizontal bars). In some experiments (8 out of the overall 72 recordings) a higher initial stimulus-evoked spiking rate was achieved using a stimulus size that spanned 20.3° in width and 16° in height of the visual field resulting in initial presentation of stimuli in 9 non-overlapping positions on the screen. In all experiments, stimulation was repeated approximately every 3–4 s, while the exact inter-stimulus interval depended on the specific down-state onset. This resulted in an inter-stimulus interval of 3.7 ± 0.1 s before pairing and 3.9 ± 0.1 s during pairing (p=0.9).

## STDP induction protocol

The STDP induction protocol was performed using custom-written routines running in Spike2 (CED; Cambridge Electronic Design) and Matlab (Mathworks, Natick, MA, USA). Before the STDP induction protocol, stable membrane potential responses were recorded for each of the four stimulus positions. Next, a stimulus position that evoked a robust and fast-onset response was selected and the average initial peak time after stimulus onset for the selected stimulus position was calculated. Then, this visual stimulus was paired with one or a few APs evoked by injection of a step current (15 ms, 1010 ± 102 pA). The time of the evoked AP was chosen so that the AP occurred within a few tens of milliseconds either before (negative timing) or after (positive timing) the time of the initial response peak. Pairing was repeated around 40 times every 3–4 s. After pairing, the visual stimuli in each of the four stimulus positions were presented for as long as the whole-cell recording lasted, with rejection of trials that did not meet the criteria for inclusion into analysis (see section 'Analysis and statistics' below). Subsequent analysis of the chosen pairing positions revealed that for both positive and negative timing experiments, the position selected for pairing ranged from the strongest to the weakest in evoked spiking responses. Overall the selection of the pairing position was not biased toward or away from the positions with the strongest pre-pairing suprathreshold responses, in that the rank of stimulus-evoked spiking at the pairing position tended neither toward first nor last among the four stimulus positions (median not significantly different from 2.5; p=0.29, n = 15 for positive timing; p=0.61, n = 17 for negative timing, Wilcoxon signed rank tests).

## Histochemistry

Rats were perfused transcardially with phosphate buffer (pH 7.2) followed by 4% paraformaldehyde in phosphate buffer. The cortex was cut coronally in 100 or 150 μm microtome sections. Biocytin-filled neurons were visualized with the avidin–biotin–horseradish peroxidase reaction (Vectastain Elite ABC kit; Vector Laboratories, Burlingame, CA, USA).

## Chemicals

Chemicals were obtained from Sigma (St. Louis, MO, USA) with the exception of Alexa Fluor 594 Na-hydrazide (Invitrogen, Carlsbad, CA, USA).

## Analysis and statistics

Membrane potential recordings were analyzed using custom written routines running in Matlab (Mathworks) and Igor (WaveMetrics, Lake Oswego, OR, USA). The down-state membrane potential preceding each stimulus was calculated as mean across a 60–70 ms 'baseline window' before the onset of the stimulus-evoked response. The peak amplitude of each stimulus-evoked subthreshold response

was calculated by determining the maximum voltage from the end of the baseline window to the end of a 'peak window', and subtracting from this the preceding down-state potential (*Figure 1C*). The peak window for each recording was determined by examining the subthreshold response averaged over all stimuli, and choosing the window to contain the first major peak in the average response (initial peak). For analysis of subthreshold stimulus-evoked responses, membrane potential responses to individual stimulus presentations were included only when they displayed no spiking from 50 ms before stimulus onset to 200 ms after and when the membrane potential remained in the down-state until the start of the response. Recordings were accepted when the amplitude of subthreshold stimulus-evoked responses before the STDP induction protocol was stable and when the resting membrane potential did not change by more than 8% during the course of the recording. For calculating the time course of membrane potential changes relative to the pairing spike (*Figure 5A–C*), we calculated the time of each neuron's pairing spikes relative to the stimulus as the maximum of a Gaussian filtered ($\sigma$ = 10 ms) peristimulus time histogram (PSTH) of APs evoked by current injection during pairing. Stimulus response centers were calculated as weighted averages using the formula: $C = \left( \sum_{1\leq i\leq 4} iR_i \right) / \left( \sum_{1\leq i\leq 4} R_i \right)$, where $R_i$ denotes the response at position $i$: the mean number of APs evoked by stimulus presentation for suprathreshold response centers, and the mean peak amplitude of the membrane potential response for subthreshold response centers. Suprathreshold response centers were calculated for neurons with the four stimulus positions arranged in a line and at least two stimulus-evoked APs before and after pairing (n = 9 neurons for positive timing, 7 for negative timing). For positive timing experiments, the total number of observed stimulus-evoked APs at the pairing position was 1.4 ± 0.9 total APs observed before induction and 9.7 ± 4.2 after (mean ± SEM across neurons, n = 15), yielding 0.06 ± 0.04 APs per stimulus before induction and 0.17 ± 0.06 after over 29.4 ± 2.1 trials before induction and 61.3 ± 9.4 after. Negative timing experiments with initial spiking yielded 29.5 ± 12.3 total APs before induction (n = 15) and 18.1 ± 6.8 after (n = 15), yielding 0.84 ± 0.23 APs per stimulus before induction and 0.66 ± 0.27 after over 36.6 ± 7.7 trials before induction and 38.4 ± 7.6 after. One-tailed t-tests were used to determine significance of sub- and suprathreshold stimulus responses at the population level, with the direction of the expected change dictated by the STDP rule. One-tailed Wilcoxon rank sum tests were used to examine individual neurons' changes in stimulus-evoked AP firing after STDP induction. Two-tailed t-tests were used to determine population wide changes in peak height and AP firing after STDP induction. The significance of average changes in the mean PSP at fixed timings before and after the pairing spike was calculated by applying one-tailed t-tests to mean voltages over time windows 60–65 ms before or 37.5–42.5 ms after the pairing spike. All other statistical evaluations of increases and decreases were two-tailed Wilcoxon rank sum tests. T-tests were used to examine association between. All reported values are mean ± SEM unless otherwise stated.

## STDP model

### Neuron model

We simulated a single postsynaptic neuron whose membrane potential $V(t)$ integrates input spikes by summing the excitatory postsynaptic potentials (EPSPs) evoked by the arrival of spikes at $N$ different synapses:

$$V(t) = \sum_{i=1}^{N} \sum_{f} w_i \varepsilon(t - t_i^f), \tag{1}$$

where $t_i^f$ is the time when the $f$-th spike arrives at synapse $i$ and $w_i$ are synapse strengths. The EPSP shape was $\varepsilon(t) = \exp(-t/\tau_{decay}) - \exp(-t/\tau_{rise})$ with $\tau_{rise}$ = 2 ms and $\tau_{decay}$ = 10 ms. Initial synapse strengths were exponentially distributed, with a mean of 2 mV chosen to reproduce the membrane potential response amplitudes observed in experimental data. The postsynaptic neuron spiked only during pairing, and at a fixed time $t_{post}$ after every stimulus.

### Plasticity

We used a simple STDP model (*Gerstner et al., 1996*; *Song et al., 2000*) where each input spike changes the synaptic strength according to a function $W$:

$$\Delta w_i = \sum_{f} W(t_{post} - t_i^f) \tag{2}$$

We used the biphasic window function

$$W(\Delta t) = A^{(+)}\theta(\Delta t)e^{-\Delta t/\tau^{(+)}} - A^{(-)}\theta(-\Delta t)e^{\Delta t/t^{(-)}}, \quad (3)$$

where $\theta(t) = 1$ for $t > 0$ and 0 otherwise, $\tau^{(+)} = \tau^{(-)} = 20$ ms, $A^{(+)} = 0.2$ and $A^{(-)} = 0.2$ for results shown in the main text and the range $0.16 \leq A^{(-)} \leq 0.24$ is explored in *Figure 5—figure supplement 3*.

## Input spike patterns
We modeled the spiking of presynaptic neurons according to three types of activity patterns:

### Time varying firing rates
Input spikes occur at random times and are generated with a time-dependent firing rate $\rho_i(t) = r_i\rho(t)$, which has the same time course $\rho(t)$ for all input neurons, but different scaling factors $r_i$ drawn from an exponential distribution (*Baddeley et al., 1997*) with mean 0.25. $\rho(t)$ is 0 until 300 ms after stimulus onset, then rises to a maximum and slowly decays (*Figure 5—figure supplement 1D*):

$$\rho^{(1)}(t > 300\text{ms}) = \rho_0\left(1 - e^{-\frac{t-300\text{ms}}{20\text{ms}}}\right)e^{-\frac{t-300\text{ms}}{3\text{s}}}, \quad (4)$$

with $\rho_0 = 5$ Hz. In mathematical terms, presynaptic spike trains are independent inhomogeneous Poisson processes.

### Patterned spike timing
Before modeling the input spikes for any stimulus presentations, a template spike train is generated separately for each presynaptic neuron using *equation 4*. The actual input spike times for a single stimulus presentation are generated by jittering the spike times in each template with Gaussian noise of mean 0 and variance $\sigma^2$. Thus spike arrival times vary across stimulus presentations with $\sigma$ controlling temporal precision. To account for unreliability in presynaptic spiking and synaptic release, each input spike crosses the synapse with probability 1/4. For these activity patterns, the mean firing rate follows a different time course for each synapse and consists of one or several Gaussian peaks (*Figure 5D–I*)

$$\rho_i(t) = \frac{1}{4}\sum_f G_{\sigma^2}(t - t)_i^f, \quad (5)$$

where $G_V$ is the probability density function for a Gaussian with mean 0 and variance $V$. The EPSPs generated at different synapses sum up to a membrane potential response with amplitude and shape similar to those observed for the time varying firing rate patterns described by *equation 4* (*Figure 5—figure supplement 2A–C*). The sole difference is a broadening of the response onset due to the spike jitter.

### Activity wave
A brief barrage of input spikes occurs shortly after stimulus onset (*Figure 5—figure supplement 1E*):

$$\rho^{(2)}(t > 300\text{ms}) = Cr_i(t - 300\text{ms})^2 e^{\frac{t-300\text{ms}}{20\text{ms}}}, \quad (6)$$

where the neuron-specific amplitudes are again drawn from an exponential distribution and $C$ is chosen so that peak firing rates are 20 Hz. The main difference from the time varying firing rate case is that inputs become active for a much shorter duration.

### Stimulus tuning of presynaptic neurons
Stimulation at position $s$ activates 400 synapses in the range $1 \leq i - 400(1 - \kappa)(s - 1) \leq 400$, where $\kappa$ defines the overlap between groups of synapses activated by different stimuli. We examined patterned spike timing inputs with 40 ms jitter for three cases: i) no synaptic overlap between stimuli ($\kappa = 0$, green trace in *Figure 5K*), ii) overlapping synaptic activations ($\kappa = 0.5$) but no shared timing, with all neurons changing their templates spike trains depending on the stimulus (blue trace in *Figure 5K*) and iii) shared synapses and shared timing, with neurons keeping the same templates for all stimuli (red trace in *Figure 5K*).

## Simulation protocol

The simulations follow a protocol similar to the experiments. We first estimated the membrane potential response before pairing by averaging over 20 stimulus presentations for each stimulus position, followed by 30 pairing trials for a single stimulus position. For time varying firing rates and patterned spike timing, the time of postsynaptic pairing spikes $t_{post}$ was either 350 ms after stimulus onset (50 ms after membrane potential response onset, n = 9 simulations) or 400 ms after stimulus onset (100 ms after membrane potential response onset, n = 15) after the onset of the membrane potential response. For the activity wave input patterns $t_{post}$ was 20 or 80 ms after response onset. These early and late values for $t_{post}$ were chosen to occur before and after the peak of the response, and the number of neurons simulated for each case was matched to experimental data. The membrane potential response after pairing was estimated by averaging over 80 trials. The simulation results plotted in the main text (*Figure 5*) were performed with patterned spike timing inputs. These plots show the difference between the membrane potential response before and after pairing for single neurons and in a population average, aligned to the time of the postsynaptic spike.

## Acknowledgements

We thank W Denk, R Froemke and Y Groemping for comments on an earlier version of this manuscript. We would also like to thank D Wallace for help with visual stimulation parameters and intrinsic imaging, J Sawinski for help with microscope modification, and N Logothetis for support.

## Additional information

### Funding

| Funder | Grant reference number | Author |
|---|---|---|
| Max Planck Society | | Verena Pawlak, David S Greenberg, Jason ND Kerr |
| Brain-i-nets | | Henning Sprekeler, Wulfram Gerstner |
| Bernstein Center for Computational Neuroscience, Tübingen, Germany | BMBF; FKZ: 01GQ1002 | Wulfram Gerstner, Jason ND Kerr |

The funders had no role in study design, data collection and interpretation, or the decision to submit the work for publication.

### Author contribution

VP, Performed all in vivo experiments and histology, Conception and design, Analysis and interpretation of data; DSG, Analysis and interpretation of data, Drafting or revising the article; HS, Devised model and performed simulations; WG, Devised model and performed simulations; JNDK, Conception and design, Analysis and interpretation of data, Drafting or revising the article.

### Ethics

Animal experimentation: All surgical procedures and experiments were conducted according to the German federal animal welfare guidelines and were approved by the animal ethics committee responsible for Tübingen, Germany (Regierungspraesidium Tübingen) under protocol numbers 3/07 and 5/09. Animals were deeply anaesthetized with Urethane (1.6–2 mg/kg), with the depth of anesthesia maintained throughout the course of the experiment with supplementary doses as required. Every attempt was made to ensure minimum discomfort to the animals at all times.

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
