## [Decision Letter]

Thank you for choosing to send your work entitled “Spike-timing dependent plasticity (STDP) changes responses of cortical neurons from sub- to suprathreshold in vivo” for consideration at *eLife*. Your article has been evaluated by a Senior Editor and 3 reviewers, one of whom is a member of *eLife's* Board of Reviewing Editors.

The Reviewing Editor and the other reviewers discussed their comments before we reached this decision, and the Reviewing Editor has assembled the following comments based on the reviewers' reports. Overall our view is that new experiments, while desirable, are not absolutely essential for the revision; however, new analysis, modeling, and revision will be required.

1. There appears to be a major discrepancy between the effect of plasticity on the subthreshold and suprathreshold responses. The changes in the subthreshold responses are quite small: for example, in Figure 2B the change in EPSP is at most 2mV, as well as in Figure 2C, the most positive pairing except two outliers show less than 20% change. Citing from the text “positive timing inductions increased the peak voltage amplitude by 21.9 ± 9.0 % (n = 15) or 1.3 ± 0.6 m”. However, in contrast the resulting change in spike probability per stimulus is huge. For positive pairing you get an increase from 0.06 sp/stim to 1.8 sp/stim (this is more than 200%). It is difficult to reconcile these two findings. The conclusion should be that the cell subthreshold response is very close to threshold such that an addition of ∼1.3 mV on average will make a 200% change in spiking probability. It would be very helpful to know not only how many cells were recorded but also how many stimulations and how many spikes were recorded. You give statistics in terms of number of neurons, but to have a reliable estimate of 0.06 spikes/stim one needs to have many stimuli, and it would be good to know how many you have recorded. From the figures, e.g. Figure 3C it looks as if there is sometimes only a single spike response in some of the conditions. The authors seem to provide evidence against the possibility that a change in Rin or depolarisation would account for it. We are not convinced that they make a very convincing case here, because they provide average data, and so it could be that on average the change in Rin is small, but that there is a good correlation between neurons that show a change in firing rate and neurons that show a change in Rin. The transition from EPSP to spike is nonlinear so the small change in Rin or in depolarisation is just as good as a candidate as a small change in EPSP size. The fact that a sham control did not show anything could be because the STDP protocol does change something, but maybe not what the authors propose. The authors therefore need to supply more analysis and provide an explanation (e.g., using modelling) how a 20% change in EPSP peak size can lead to a 200% in spike probability.

2. Related to the previous point, a surprising finding is that the LTD protocol strongly affects subthreshold peak amplitudes (in fact, the effect is more robust as compared to the LTP protocol, -32.4% vs +21.9%), but it does not seem to affect spiking output, whereas the LTP paradigm does. Perhaps this is caused by the large spread and not-normal distribution of baseline spiking within the cell population that was used for the LTD protocol, and perhaps by the relatively low number of cells. Here the authors should have probably aimed for applying the protocol to robustly spiking neurons only. Clearly, a decrease in spiking will be hard to detect in cells that barely spike to start with. Right now the data look more like a negative control for the LTP experiments. As the paper does not further address this finding, we think the authors should more clearly discuss this and analyze the data to address this apparent discrepancy – e.g., why could the experiments be inconclusive; or, if the authors believe the experiments are conclusive, what would then be the functional/behavioral relevance of LTD?

3. It seems that the effects of STDP on sub and suprathreshold activity were analyzed independently. The authors must have information on a possible relationship within cells. Does LTD/LTP paradigm affect the gain and threshold of the neuronal input – output function? (e.g., see Carvalho and Buonomano, *Neuron*, 2009)?

4. No negative control experiments are provided in the text or figures (at least regarding subthreshold activity). Is LTP/LTD really dependent on spike pairing? In other words, could the effect depend on just depolarization instead? Similarly, since the pairing was performed at different positions there is likely a great variability in pre-pairing PSP peak amplitudes. The authors may want to check whether or not the success rate or level of STDP related to pre-peak amplitudes, and provide a scatterplot of pre vs post pairing EPSP amplitudes.

5. The relevance of Figure 4 is unclear. There are no statistics provided – are the results in Figure 4C significant or not? In Figure 4C, we are told there are shifts, but what are these shifts and what do they signify? Why is this “spatial reorganization” (this and other figures)? The Distance from Paired Position: is that the same as the Position from Paired Position and the Distance from Pairing Position (Figure 3)? Is this really a true distance because it is in a part of the RF? In the text the authors use this figure to essentially describe a general, qualitative phenomenon for which the statistics are not provided, and then they elaborate on it as if it is proven. Either the authors should use statistical tests to confirm the significance of the results; or they should provide more data; or they should remove this part of the paper.

6. The neuromodulation results (Figure 6) are not obviously related to the rest of the study, and the experiments are of lower impact and quality than those in the rest of the paper. Although the role of neuromodulation in learning and synaptic plasticity is important, the link with the rest of the paper is not well explained, and Figure 6 therefore seems out of context. What does the reader learn from the contents of Figure 6 that is critical to their understanding of the other results? To know that STDP rules change the tuning cortical neuron spiking, why does the reader need to learn about the need for cholinergic neuromodulation in particular? Moreover, the biological relevance of cholinergic blockade is unclear – i.e., why does in vivo plasticity require cholinergic neuromodulation, when in vitro plasticity clearly does not (e.g., Markram et al *Science* 1997; Froemke *Nature* 2002; Feldman *Neuron* 2000; Sjostrom et al *Neuron* 2001 to provide few examples). Given these concerns, we recommend removing these results and the corresponding figure from the paper as it currently does not significantly strengthen the manuscript.

7. The authors use the STDP model as the explanation for why they observe this increase before APs (LTP window) and a decrease after (LTD window). But as the authors do not test the LTD window experimentally, this decrease may in principle be due to something else, such as increase in inhibition, a change in I_H, or some other intrinsic conductance. In general, the authors have pushed very strongly the idea that STDP in particular (as opposed to classical Hebbian LTP/LTD) underlies the phenomenology. Certainly the results are consistent with STDP, but they do not demonstrate definitively that STDP definitively underlies the plasticity. For example, Blum and Abbott's Neural Comput 1996 paper was based on classical Hebbian learning, yet has in hindsight been reinterpreted as supporting STDP, but already the Hebbian postulate itself specifies the need for temporal order and temporal asymmetry in in learning rules: cell A has to repeatedly and persistently fire before cell B. And the LTD could in principle be due to Gunther Stent's LTD. So why do we need STDP to explain this? The fact that Pawlak et al use a single spike to induce plasticity does not necessarily mean this is STDP; it just means a single postsynaptic spike is enough. The alternative possibilities need to be acknowledged and discussed more thoroughly.

---

## [Author Response]

*Overall our view is that new experiments, while desirable, are not absolutely essential for the revision; however, new analysis, modeling, and revision will be required*.

We thank the reviewers and reviewing editor for supplying a clear and constructive review of our manuscript. We have addressed the concerns by performing additional experiments and carrying out further analysis. Although the editors did not recommend experiments, we felt that the reviewers raised some very good points that could be addressed experimentally. This combined with new analysis, and a rewritten introduction and discussion, now allows the manuscript to make clearer and stronger statements. With the new experiments, we now show that STDP rules *in vivo* can convert a neuron's visually evoked responses from non-spiking to spiking, can increase and diminish existing spiking responses, and can induce a rearrangement of the neurons sub- and suprathreshold responses to stimuli presented at locations surrounding the paired location. We have rewritten most of the introduction to realign the findings more towards a general plasticity rule. We have also rewritten almost the entire discussion to address reviewers' questions. Using the reviewers' comments we have tightened up the argument and logical flow.

*1. There appears to be a major discrepancy between the effect of plasticity on the subthreshold and suprathreshold responses. The changes in the subthreshold responses are quite small: for example, in Figure 2B the change in EPSP is at most 2mV, as well as in Figure 2C, the most positive pairing except two outliers show less than 20% change. Citing from the text “positive timing inductions increased the peak voltage amplitude by 21.9* ± *9.0 % (n = 15) or 1.3* ± *0.6 m”. However, in contrast the resulting change in spike probability per stimulus is huge. For positive pairing you get an increase from 0.06 sp/stim to 1.8 sp/stim (this is more than 200%). It is difficult to reconcile these two findings. The conclusion should be that the cell subthreshold response is very close to threshold such that an addition of ∼1.3 mV on average will make a 200% change in spiking probability*.

This is a good point, but we do not think there is a discrepancy between these findings as it is not clear what voltage changes would be expected at the soma to change spiking. Relating synaptically derived voltage changes at the soma to the binary output spiking changes is very complex and somewhat unknown, especially *in vivo* due to the combinations of various non-linear currents over the sub-threshold voltage range and the spatial location of the evoked synaptic inputs relative to the soma *in vivo* during visual stimulation. Given the non-linear rectification of the current-voltage relationship near threshold, the strong attenuation of synaptic events recorded at the soma, and the variable spike threshold recorded at the soma, it is not clear to us what the expected voltage changes at the soma should be to elicit spiking changes in the neuron. We have now addressed this in the discussion, results, and introduction.

In addition, with our new experiments we now show that with negative timing stimulus evoked spiking responses decrease compared to pre-paring periods. This was also associated with a small hyperpolarizing change in the membrane potential of the same magnitude as the depolarizing change associated with the positive paring protocol. In some respects this is the driving force of the question: whether the small changes in membrane potential brought about by STDP protocols can lead to functional changes at the spiking level.

*It would be very helpful to know not only how many cells were recorded but also how many stimulations and how many spikes were recorded. You give statistics in terms of number of neurons, but to have a reliable estimate of 0.06 spikes/stim one needs to have many stimuli, and it would be good to know how many you have recorded. From the figures, e.g. Figure 3C it looks as if there is sometimes only a single spike response in some of the conditions*.

We have added more information about the number of presented stimuli and stimulus-evoked APs as suggested. While the significant increases in spiking at the population level for positive (LTP) and negative (LTP) timings remain statistically valid regardless of the number of stimulus presentations and spikes available, we agree that an analysis that takes into account the amount of data available for each neuron is desirable as well. We have therefore included significance tests for individual neurons, revealing that 4 neurons for positive timings and another 4 for negative timings show significant changes in AP firing. Together, these findings show that our sampling is sufficient to detect changes in AP firing.

*The authors seem to provide evidence against the possibility that a change in Rin or depolarisation would account for it. We are not convinced that they make a very convincing case here, because they provide average data, and so it could be that on average the change in Rin is small, but that there is a good correlation between neurons that show a change in firing rate and neurons that show a change in Rin*.

We have performed the analysis as suggested revealing that neither changes in Rin nor changes in resting membrane potential are correlated to changes in stimulus evoked AP-firing.

*The transition from EPSP to spike is nonlinear so the small change in Rin or in depolarisation is just as good as a candidate as a small change in EPSP size. The fact that a sham control did not show anything could be because the STDP protocol does change something, but maybe not what the authors propose. The authors therefore need to supply more analysis and provide an explanation (e.g., using modelling) how a 20% change in EPSP peak size can lead to a 200% in spike probability*.

Recent evidence (Tchumatchenko, Malyshev et al. 2011) has shown that cortical pyramidal neurons are able to rapidly and reliably change firing rates in response to very small changes in depolarization (2-fold spiking difference induced with a 40pA current injection) and hyperpolarization. Relating synaptically derived voltage changes at the soma to the binary output spiking changes is very complex, and somewhat unknown, especially *in vivo* due to the combinations of various non-linear currents over the sub-threshold voltage range (Waters and Helmchen 2006) and spatial complexity of evoked synaptic inputs *in vivo* during visual stimulation. Given the non-linear rectification of the current-voltage relationship near threshold, the strong attenuation of synaptic events recorded at the soma, the precision with which neurons can follow complex membrane-potential waveforms at the soma to reliably generate spikes (Mainen and Sejnowski 1995) and finally, the variable spike threshold recorded at the soma (Azouz and Gray 2000), it is not clear to us what the expected voltage changes at the soma should be to elicit spiking changes in the neuron. We also note that the observed increase in spiking displayed stimulus specificity (Figure 3G), but this would not be observed if the increase were merely due to changes in Rin or resting membrane potential.

*2. Related to the previous point, a surprising finding is that the LTD protocol strongly affects subthreshold peak amplitudes (in fact, the effect is more robust as compared to the LTP protocol, -32.4% vs +21.9%), but it does not seem to affect spiking output, whereas the LTP paradigm does. Perhaps this is caused by the large spread and not-normal distribution of baseline spiking within the cell population that was used for the LTD protocol, and perhaps by the relatively low number of cells. Here the authors should have probably aimed for applying the protocol to robustly spiking neurons only. Clearly, a decrease in spiking will be hard to detect in cells that barely spike to start with. Right now the data look more like a negative control for the LTP experiments. As the paper does not further address this finding, we think the authors should more clearly discuss this and analyze the data to address this apparent discrepancy – e.g., why could the experiments be inconclusive; or, if the authors believe the experiments are conclusive, what would then be the functional/behavioral relevance of LTD*?

In our minds this was a very serious concern and undermined the strength of the study; in fact, this concern was the driving force behind performing new experiments. What we did not want was a finding that was based on low numbers of recordings, and wrong for this reason. With this in mind we added additional experiments and now show that STDP rules can both increase and decrease stimulus evoked AP firing. All figures have been updated to reflect this finding.

*3. It seems that the effects of STDP on sub and suprathreshold activity were analyzed independently. The authors must have information on a possible relationship within cells*.

No significant correlation was observed between subthreshold and suprathreshold changes at the pairing position (see the following scatter plot: Author response image 1). Correlation was not significant (p > 0.05, t-tests) when considering positive timing experiments, negative timing experiments, or all experiments. A clear relationship between subthreshold and suprathreshold changes was not expected, however.

*Does LTD/LTP paradigm affect the gain and threshold of the neuronal input – output function? (e.g., see Carvalho and Buonomano, Neuron, 2009)*?

While estimating the effect of plasticity change on the I/O relationship would of course be of great interest, we feel that it would be another study in itself and without having available the careful *in vitro* manipulations that were available to Carvalho and Buonomano (2009) for our *in vivo* study, we would not be able to add new or insightful data to the present study. One of the key features of the Carvalho and Buonomano (2009) study was that they were able to manipulate neuronal properties using pharmacology and isolate certain currents and voltages that they later implemented in their models. The accuracy needed to do this correctly is not afforded *in vivo*, yet, and would be a tour-de-force technically.

*4. No negative control experiments are provided in the text or figures (at least regarding subthreshold activity). Is LTP/LTD really dependent on spike pairing? In other words, could the effect depend on just depolarization instead*?

Both the spiking responses and the sub-threshold responses to visual stimulation over extended periods of time do not significantly change – either decreasing or increasing – without the addition of deliberately placed pairing spikes. It is not spikes alone, but the pairing of the spike with the visual stimulation evoked pre-synaptic avalanche. The on-going spiking, that is not associated with the stimulus, does not produce the same predictable changes in either sub- or suprathreshold voltage responses.

*Regarding the back-propagating APs vs. sharp brief depolarization of the soma without generating the back-propagating AP, it is hard to imagine that the depolarizing current would spread far from the recording site (soma) into the dendrites and, ultimately the spines activated from the visual stimulation*.

Similarly, since the pairing was performed at different positions there is likely a great variability in pre-pairing PSP peak amplitudes. The authors may want to check whether or not the success rate or level of STDP related to pre-peak amplitudes, and provide a scatterplot of pre vs post pairing EPSP amplitudes.

The percentage changes in peak amplitude we observed at the pairing position were not correlated with the initial peak amplitude before pairing for either positive or negative timing experiments (p > 0.05, t-tests). As shown here (Author response image 2), while variability across experiments was observed for both positive and negative timing experiments, the amplitude of these changes was not determined by the initial peak height.

The suggested scatter plot of peak amplitudes before and after plasticity induction reveals (shown below: Author response image 3) reveals the same lack of dependence.

Thus the plasticity effects observed are not restricted to neurons with large or small initial peaks.

*5. The relevance of Figure 4 is unclear. There are no statistics provided – are the results in Figure 4C significant or not? In Figure 4C, we are told there are shifts, but what are these shifts and what do they signify*?

There are appropriate statistics provided. We have tested the significance by examining the correlation between the shift in the response center and the distance from the response center to the induction position. Positive timing experiments showed a significant tendency for the response center to move toward the pairing position, while negative timing experiments did not show any significant trend. In the revised manuscript, the statistical analysis of suprathreshold response center shifts now appears as such: “Stimulus response centers were calculated as weighted averages of the four stimulus positions…”

Statistical analysis of subthreshold response center shifts now appears as such: “… We next examined whether the shifts in response centers observed for suprathreshold responses were also reflected in subthreshold responses…”

*Why is this “spatial reorganization” (this and other figures)*?

The response center describes the dependence of sub- or suprathreshold responses on the spatial positioning of the stimulus on a screen in front of the animal, taking into account responses to stimuli at all positions. Thus a shift in the response center reflects changes in the relative amplitude of sub- or suprathreshold responses across the space of locations used for stimulus presentation, a process we referred to as spatial reorganization.

*The Distance from Paired Position: is that the same as the Position from Paired Position and the Distance from Pairing Position (Figure 3)*?

We have corrected this instance of inconsistent terminology, and normalized the nomenclature used to describe sensory stimulation throughout the paper. We now use the phrase “distance from paired position”.

*Is this really a true distance because it is in a part of the RF? In the text the authors use this figure to essentially describe a general, qualitative phenomenon for which the statistics are not provided, and then they elaborate on it as if it is proven. Either the authors should use statistical tests to confirm the significance of the results; or they should provide more data; or they should remove this part of the paper*.

This was never used to quantify a qualitative measure, as the strength of the effect (R value) as well as the significance (P value) have been quantified. What we clearly show is the rearrangement neuronal responses based on STDP rules and the spatial dependence of these response both at the supra and subthreshold level.

*6. The neuromodulation results (Figure 6) are not obviously related to the rest of the study, and the experiments are of lower impact and quality than those in the rest of the paper. Although the role of neuromodulation in learning and synaptic plasticity is important, the link with the rest of the paper is not well explained, and Figure 6 therefore seems out of context. […] Given these concerns, we recommend removing these results and the corresponding figure from the paper as it currently does not significantly strengthen the manuscript*.

We have followed these suggestions and removed the text and figures describing neuromodulatory effects from the manuscript.

*7. The authors use the STDP model as the explanation for why they observe this increase before APs (LTP window) and a decrease after (LTD window). But as the authors do not test the LTD window experimentally, this decrease may in principle be due to something else, such as increase in inhibition, a change in I_H, or some other intrinsic conductance. […] The alternative possibilities need to be acknowledged and discussed more thoroughly*.

We thank the reviewers for this comment and we have devoted a paragraph of our discussion to this point.